# Protein Phosphatase 4 Is Required for Centrobin Function in DNA Damage Repair

**DOI:** 10.3390/cells12182219

**Published:** 2023-09-06

**Authors:** Zsuzsánna Réthi-Nagy, Edit Ábrahám, Rita Sinka, Szilvia Juhász, Zoltán Lipinszki

**Affiliations:** 1MTA SZBK Lendület Laboratory of Cell Cycle Regulation, Institute of Biochemistry, HUN-REN Biological Research Centre, H-6726 Szeged, Hungary; nagy.zsuzsanna@brc.hu (Z.R.-N.); abraham.edit@brc.hu (E.Á.); 2Doctoral School of Biology, Faculty of Science and Informatics, University of Szeged, H-6726 Szeged, Hungary; 3National Laboratory for Biotechnology, Institute of Genetics, HUN-REN Biological Research Centre, H-6726 Szeged, Hungary; 4Department of Genetics, University of Szeged, H-6726 Szeged, Hungary; rsinka@bio.u-szeged.hu; 5Institute of Biochemistry, HUN-REN Biological Research Centre, H-6726 Szeged, Hungary

**Keywords:** protein phosphatase 4, Centrobin, DNA damage response, homologous recombination repair, arms-closed chromosome morphology, DNA double-stranded break

## Abstract

Genome stability in human cells relies on the efficient repair of double-stranded DNA breaks, which is mainly achieved by homologous recombination (HR). Among the regulators of various cellular functions, Protein phosphatase 4 (PP4) plays a pivotal role in coordinating cellular response to DNA damage. Meanwhile, Centrobin (CNTRB), initially recognized for its association with centrosomal function and microtubule dynamics, has sparked interest due to its potential contribution to DNA repair processes. In this study, we investigate the involvement of PP4 and its interaction with CNTRB in HR-mediated DNA repair in human cells. Employing a range of experimental strategies, we investigate the physical interaction between PP4 and CNTRB and shed light on the importance of two specific motifs in CNTRB, the PP4-binding FRVP and the ATR kinase recognition SQ sequences, in the DNA repair process. Moreover, we examine cells depleted of PP4 or CNTRB and cells harboring FRVP and SQ mutations in CNTRB, which result in similar abnormal chromosome morphologies. This phenomenon likely results from the impaired resolution of Holliday junctions, which serve as crucial intermediates in HR. Taken together, our results provide new insights into the intricate mechanisms of PP4 and CNTRB-regulated HR repair and their interrelation.

## 1. Introduction

The DNA damage response (DDR) is a surveillance system evolved to safeguard genome integrity in the face of genotoxic stress throughout an organism’s life. Upon DNA damage, a complex signaling pathway activates the DDR to coordinate the detection of DNA lesions (checkpoint activation), cell cycle arrest, and efficient DNA repair [1]. Following successful repair, DDR activates checkpoint recovery to resume cell cycle progression [2,3]. One of the most severe forms of DNA damage is double-stranded breaks (DSBs), which require immediate repair through processes such as homologous recombination (HR) or non-homologous end joining (NHEJ) [4]. Dysregulation of these processes may produce gene mutations, chromosome aberration, and aneuploidy [5], hallmarks of proliferative diseases and genetic disorders [6,7]. Therefore, precise control and fidelity are crucial during the repair process.

Reversible protein phosphorylation is a highly conserved molecular mechanism in eukaryotes that orchestrates complex cellular processes, including cell cycle progression and DNA damage response [8,9]. Phosphorylation is catalyzed by protein kinases and reversed by protein phosphatases, and serves as a molecular switch, modulating the activity, structure, half-life, localization, and interacting partners of target proteins. Checkpoint activation and DNA repair are mainly regulated by the extensively studied master kinases, the Ataxia–telangiectasia and Rad3-related (ATR), Ataxia–telangiectasia mutated (ATM), and DNA-dependent protein kinases (DNA-PK) [10], as well as their effectors, Checkpoint kinase 1 (Chk1) and Checkpoint kinase 2/Radiation-sensitive 53 (Chk2/RAD53, mammal/yeast nomenclature) [11]. However, dynamic phosphorylation and dephosphorylation of DDR regulators and executor molecules are critical for successful DNA repair and maintenance of genome integrity. It has become evident over the past years that several protein phosphatases that counteract the activity of DDR kinases and dephosphorylate their substrates are involved in this process. The five major phosphatases that contribute directly to DDR regulation are the Ser/Thr Phosphoprotein phosphatase 1 (PPP1), Phosphoprotein phosphatase 2A (PPP2A), Phosphoprotein phosphatase 2C/WIP1 and Phosphoprotein phosphatase 4 (PPP4, hereafter PP4), as well as Cdc14, which is a dual specificity enzyme [12,13].

PP4 is a ubiquitous and essential phosphatase that regulates a variety of cellular processes, including centrosome maturation and microtubule dynamics, cell division, development, and differentiation, as well as DDR (reviewed in [14,15]). It has been reported that PP4 dephosphorylates several key proteins during DDR, such as ATR kinase, RAD53/Chk2 kinase, γH2AX histone, Replication protein 2A (RPA2), p53-binding protein 1 (53BP1), Deleted in breast cancer 1 protein (DBC1), and Kinesin-associated protein-1 (KAP-1) [16,17,18,19,20,21,22]. These findings have shed light on the necessity of PP4 in all stages of DDR, including checkpoint activation and double-stranded DNA break repair by HR, as well as NHEJ. PP4 is believed to function primarily by promoting cell recovery and the resumption of the cell cycle after successful repair [12,13,15].

The major form of PP4, which is present from yeast to humans, is the heterotrimeric complex consisting of an evolutionarily conserved catalytic subunit (PP4c), a scaffolding subunit (PPP4R2) and a regulatory subunit (PPP4R3, hereafter R3) [14,23,24]. The R3 subunit (Psy2 in yeast, Falafel in *Drosophila*, and PPP4R3A (hereafter R3A) and PPP4R3B (hereafter R3B) isoforms in mammals) is responsible for subcellular localization, as well as substrate recognition of PP4 [19,24,25,26,27,28,29,30,31]. We and others have shown that the R3 subunit, through its non-canonical amino-terminal EVH1 domain, specifically binds to conserved short linear motifs (SLiMs: FxxP or MxPP, where x can be any amino acid) in the target proteins [24,29,30]. Substitution of phenylalanine (F) and proline (P) in FxxP, or methionine and both prolines (PP) in MxPP to alanine (i.e., AxxA or AxAA, respectively), completely abolished the interaction between PP4 and its substrates [29,30]. Interestingly, these studies in *Drosophila* and human cells have identified novel FxxP/MxPP-containing DDR targets of PP4, including the centrosomal BRCA2-interacting protein (Centrobin) [29,30].

Centrobin (CNTRB) was initially identified as a centrosomal protein that interacts with BRCA2 in human cells [32]. It plays a vital role in daughter centriole duplication and elongation [32,33], and contributes to microtubule stability and dynamics, regulated by NEK2 and polo-like kinase 1 [34,35,36,37,38]. CNTRB has been identified as a putative substrate of ATM/ATR kinases, which recognize and phosphorylate specific motifs, such as serine-glutamine (SQ) or threonine–glutamine (TQ) [39]. Recent studies have demonstrated that ATR kinase specifically phosphorylates CNTRB in response to UV-induced DNA damage, leading to its enrichment in the nuclear matrix. This modification is essential for cell survival and proper DNA repair through HR [40]. However, the identity of the phosphatase responsible for dephosphorylating CNTRB in DDR remains unknown. We note that the previously identified PP4-binding motif in human CNTRB (771-FRVP-774 [29]) is in close proximity to the putative ATR-targeted SQ motif of CNTRB (781-SQ-782, Figure 1a). In addition, it is known that PP4 interacts with the ATR-interacting protein, ATRIP and, together with ATR, they co-regulate the phospho-status of different substrates [22]. This leads to the speculation that PP4 may play a role in controlling the function of CNTRB in DNA repair.

In this study, we provide evidence for the physical interaction between PP4 and CNTRB. We also show that the integrity of the FRVP and SQ motifs in CNTRB is crucial for repairing irradiation-induced double-stranded DNA breaks through HR in human cells. Exclusion of PP4 from CNTRB or mutation of the SQ motif results in a significant delay in HR and the accumulation of chromosomes with abnormal morphology, likely due to defective Holliday junction resolution [5,41]. Our findings imply a novel function of PP4 in the late stages of HR, and further support the role of PP4 in facilitating cell cycle resumption following DNA repair.

## 2. Materials and Methods

### 2.1. DNA Constructs

cDNA-encoding human PPP4R3A isoform 1 (UniProt ID: Q6IN85; cDNA name: MHS6278-202759611, clone ID: 6142109), human PPP4R3B isoform 3 (UniProt ID: Q5MIZ7-3, cDNA name: MHS6278-202807853; clone ID: 5259789) and human Centrobin isoform 1 (UniProt ID: Q8N137-1; cDNA name: MHS6278-202832427, clone ID: 4859539) were originally generated by the Mammalian Gene Collection project [42] and we obtained from Horizon Discovery Ltd. (Cambridge, UK). Coding DNA sequence (CDS) of R3A and R3B were cloned into the pDONR221 plasmid by BP reaction using the Gateway System (cat#12536017, Thermo Fisher Scientific, Waltham, MA, USA). Expression constructs were made by LR reaction using the following destination vectors: pDEST15 (for N-terminal GST-fusion in *E. coli*, cat#11802014, Thermo Fisher Scientific, Waltham, MA, USA) and pcDNA-DEST53 (for N-terminal GFP fusion in mammalian cells, cat#12288015, Thermo Fisher Scientific, Waltham, MA, USA). Modified forms of CNTRB (CNTRB-ARVA (771-FRVP-774 to 771-ARVA-774), S781A or S781D) were created by standard mutagenesis using QuickChange II XL Site-Directed Mutagenesis Kit (cat#200522, Agilent Technologies, Santa Clara, CA, USA). CDS of full-length CNTRB, CNTRB-ARVA, CNTRB-S781A and CNTRB-S781D were subcloned into the pHY22 plasmid (for in vitro expression [43]) by standard procedure. Truncated forms of CNTRB (1-180aa, 180-903aa, 1-450aa, 450-903aa) were generated by PCR and cloned into the pFlag-CMV4 plasmid (for N-terminal Flag fusion in mammalian cells, cat#E7158, Merck Millipore, Burlington, MA, USA). CDS of ARVA, S781A, and S781D mutants of CNTRB^180-903aa^ were also cloned into the pFlag-CMV4 plasmid. All DNA constructs were verified by DNA sequencing. Oligonucleotide primers are provided in Appendix A.

### 2.2. Recombinant Protein Expression and Purification

Glutathione S-transferase (GST) fused recombinant full-length R3A (GST-R3A) and R3B (GST-R3B) were expressed in six-pack *E. coli* cells [44]. Bacteria cultures were grown at 16 °C for 48 h (at 280 r.p.m. using an orbital shaker with 19 mm diameter) in Terrific broth autoinduction media (cat#AIMTB0210, Formedium, Hunstanton, UK), then bacteria were harvested by centrifugation (3500*× g*, 4 °C, 15 min) and resuspended in ice-cold phosphate-buffered saline (PBS) supplemented with 0.2 mg/mL lysozyme (cat#L6879, Sigma-Aldrich, St. Louis, MO, USA) and 1 mM phenylmethylsulfonyl fluoride (PMSF, cat#P7626, Sigma-Aldrich, St. Louis, MO, USA). Cells were lysed by standard sonication followed by centrifugation at 12,000*× g*, 4 °C, 20 min. Recombinant proteins were affinity purified on glutathione sepharose 4B resin according to the manufacturer (cat#17-0756-01, Cytiva, Washington, WA, USA). Immobilized bait proteins on beads (GST, GST-R3A or GST-R3B, respectively) were stored in 50% glycerol in PBS at −20 °C.

### 2.3. In Vitro Pull-Down Assay

To test the direct interaction between CNTRB (prey) or its derivatives (CNTRB-ARVA, CNTRB-S718A, and CNTRB-S718D) and GST-R3A or GST-R3B (bait) protein, respectively, we performed a GST-IVTT assay as described earlier [43]. Briefly, ^35^S-methionine-labeled CNTRB or its derivatives were produced in coupled in vitro transcription and translation reaction (IVTT, at 30 °C for 1 h) using the TNT Quick Coupled Transcription/Translation System (cat#L1170, Promega, Madison, WI, USA). Prey proteins were diluted in binding buffer (50 mM HEPES pH 7.5, 150 mM NaCl, 2 mM MgCl_2_, 1 mM EGTA, 1 mM DTT, 0.1% Triton X-100, EDTA-free protease inhibitor cocktail (PIC, cat#11873580001, Roche, Basel, Switzerland) and 0.5% standard bovine serum albumin, incubated with equal amounts of immobilized GST, GST-R3A, or GST-R3B proteins, respectively, and mixed at 4 °C for 2 h with gentle rotation. Beads were settled by centrifugation (500*× g*, 4 °C, 5 min), washed three times (5 min each) with washing buffer (50 mM HEPES pH 7.5, 150 mM NaCl, 2 mM MgCl_2_, 1 mM EGTA, 1 mM DTT, 0.1% Triton X-100) and four times (5 min each) with washing buffer supplemented with 50 mM NaCl and 0.1% Triton X-100. Beads were boiled in Laemmli sample buffer for 4 min, IVTT inputs and eluted proteins were run on SDS-PAGE, gels were stained with Coomassie brilliant blue, scanned, dried, and subjected to autoradiography.

### 2.4. Cell Culture Maintenance

In this study all cell lines were cultured in Dulbecco’s modified Eagle’s medium (DMEM) containing GlutaMAX™ Supplement (cat#61965026, Thermo Fisher Scientific, Waltham, MA, USA) supplemented with 10% fetal bovine serum (FBS, cat#ECS0180L, Euroclone, Pero, Italy), 1xPenStrep (cat#XC-A4122, Biosera, Nuaille, France), and 1% nonessential amino acid (NEAA, cat#BE13-114E, Lonza, Basel, Switzerland) and maintained at 37 °C under 5% CO_2_, in a humidified incubator.

### 2.5. Co-Immunoprecipitation

Human HEK293 cells (ATCC CRL-1573, Manassas, VA, USA) were transiently co-transfected with GFP, GFP-R3A or GFP-R3B and Flag-tagged CNTRB^180-903aa^ and its ARVA, S781A or S781D mutants, respectively, using polyethylenimine (PEI, cat#408727, Merck Millipore, Burlington, MA, USA), according to standard procedures. Cells were harvested 2 days post-transfection and lysed in Buffer A (50 mM Tris pH 7.6, 50 mM NaCl, 2 mM MgCl_2_, 0.5 mM EGTA, 0.1% NP-40, 5% glycerol, 1 mM DTT, 1 mM PMSF, 1x PIC, 25 µM MG132 (cat#10012628, Cayman Chemical Company, MI, USA) and 0.1 µL/mL Benzonase nuclease (cat#70746-10KUN, Merck Millipore, Burlington, MA, USA) by passing the cell suspension through a G25 needle ten times. Lysates were centrifuged (17,000× *g*, 20 min, 4 °C) and cleared supernatants were used for immunoblotting (inputs) and co-immunoprecipitations (co-IP). For co-IP clarified lysates were mixed with GFP-trap magnetic agarose beads (cat#gtma-20, ChromoTek GmbH, Planegg, Germany) for 90 min at 4 °C, with gentle rotation. Beads were washed four times (5 min each) in Buffer B (50 mM Tris pH 7.6, 50 mM NaCl, 2 mM MgCl_2_, 0.5 mM EGTA, 0.1% NP-40, 5% glycerol), proteins were eluted by boiling in Laemmli sample buffer and analyzed by SDS-PAGE followed by immunoblotting using anti-FlagM_2_ and anti-GFP antibodies, respectively.

### 2.6. Gene Silencing, Western Blotting, and qPCR

In HeLa (ATCC CRM-CCL-2, Manassas, VA, USA) or DR-GFP reporter U2OS [45] cells CNTRB, PP4c, R3A, and R3B were silenced separately, or CNTRB and PP4c we co-silenced using the following two sets of siRNAs: CNTRB (cat#4392420 siRNA ID #1: 141226, cat#AM16708 siRNA ID #2: s42056), R3A (cat#AM16708 siRNA ID #1: s31226, cat#4392420 siRNA ID #2: 133612), R3B (cat#AM16708 siRNA ID #1: s32915, cat#4392420 siRNA ID #2: 123224), PP4c (cat#AM16708 siRNA ID #1: s10999, cat#4390824 siRNA ID #2 4441), and Control (cat#4390843). All siRNAs were purchased from Thermo Fisher Scientific (Waltham, MA, USA). Transfection of cell lines with specific siRNAs was carried out using Lipofectamine (cat#11668-018, Invitrogen, Waltham, MA, USA) or DharmaFECT^TM^ (cat#T-2022-02, GE Healthcare Dharmacon, Inc., Lafayette, CO, USA) transfection reagent according to the manufacturer’s protocol. Experiments were performed 48 h after siRNA transfection.

Total protein extracts were made by boiling the cells in Laemmli sample buffer for 5 min; then, 10 µg of each protein sample was run on SDS-PAGE followed by immunoblotting using anti-CNTRB, anti-PP4c, anti-R3A, anti-R3B, and anti-αTubulin antibodies, respectively.

Total RNA purification was performed with Quick-RNA MiniPrep kit (cat#R1054, Zymo Research, Irvine, CA, USA). For the synthesis of first-strand cDNA, the RevertAid First Strand cDNA Synthesis Kit (cat#K1670, Thermo Fisher Scientific, Waltham, MA, USA) was used according to the manufacturer’s instructions. Maxima SYBR Green/ROX qPCR Master Mix (cat#K0222, Thermo Fisher Scientific, Waltham, MA, USA) was used for the real-time quantitative PCR reaction, according to the manufacturer’s instructions. Reactions were run three times in quadruplicates in the Rotor-Gene Q Real-Time PCR Detection System (QIAGEN, Germantown, MD, USA) with the following reaction conditions: 95 °C 10 min, 40 cycles of 95 °C 15 s, 55 °C 30 s, 72 °C 30 s. The final values represent the mean and standard error of the quadruplicates.

### 2.7. γH2AX Foci Quantitation Assay

HeLa cells were cultivated in 6-well tissue culture plates (cat#30006, SPL Life Sciences, Pochon, Kyonggi-do, South Korea), and following 24 h of gene silencing, the cells were trypsinized (cat#TRY-3B, Capricorn Scientific, Ebsdorfergrund, Germany) using standard protocols and transferred to glass-bottom cell culture chambers (cat#631-0150, VWR International, Radnor, PA, USA). Twenty-four hours post gene silencing, the cells were exposed to 2 Gy X-rays using a Trakis XR-11 X-ray machine according to previous work [46]. At various time points after irradiation, the cells were fixed with 3% PFA (paraformaldehyde, cat#158127, Merck Millipore, Burlington, MA, USA) for 10 min and subjected to immunofluorescence. The γH2AX foci were quantified in 50 EdU-positive S-phase cells after anti-γH2AX immunostaining. The remaining cells were collected for qPCR analysis.

### 2.8. EdU Assay

For the specific labeling of S-phase cells, a medium containing 10 µM 5-ethynyl 2′-deoxyuridine (EdU, cat#BCN-001-5, BaseClick GmbH, Munich, Germany) was added to the cells 1 h before X-ray irradiation. At various time points, the cells were fixed with 3% PFA for 10 min and subjected to anti-γH2AX staining, as described below. To label the EdU positive cells, Edu-Click 555 (cat#BCK-EdU555-1, BaseClick GmbH, Munich, Germany) was used, following the manufacturer’s protocol. In the end, nuclei were counterstained with 1 μg/mL Hoechst33342 dye (cat#H21492, Thermo Fisher Scientific, Waltham, MA, USA) in PBS. Fluorescence excitation was performed using diode laser 405, 488, and 555 nm with LSM800 confocal microscope (Carl Zeiss, Jena, Germany).

### 2.9. Statistical Analysis of Data

All experiments were replicated at least three times. Statistical analysis was performed using Origin Graph statistical software (https://www.originlab.com/origin (accessed between 1 March 2023 and 1 August 2023)). In all cases, we made an ANOVA test [47]. Asterisks represent *p* values, which correspond to the significance of regression coefficients (* *p* < 0.05, ** *p* < 0.01, and *** *p* < 0.001).

### 2.10. DR-GFP Reporter Assay

DR-GFP U2OS cells, obtained from Jeremy Stark [45], were initially seeded in 6-well tissue culture plates, followed by the silencing of target genes. After 24 h, the cells were trypsinized and transferred to glass-bottom cell culture chambers. Subsequently, 24 h post-transfection, the cells were transfected with an I-SceI rare cutter endonuclease coding plasmid construct (cat#26477, Addgene Watertown, MA, USA). After 96 h of gene silencing, the cells were fixed with 3% PFA for 10 min. The cells were then visualized using confocal microscopy, and the number of GFP-positive cells, indicative of homologous recombination (HR) events, was quantified using CellProfiler software (version 4.2.6). In CellProfiler, the nuclei were segmented based on Hoechst-stained DNA. A minimum of 3000 cells were analyzed per experimental condition (Appendix A).

### 2.11. Chromosome Preparation

HeLa cells were seeded in 6-well tissue culture plates and transfected with siRNAs, or co-transfected with siRNAs and flag constructs. After 24 h, cells were separated into two 6-well plates, one was irradiated with X-ray using Trakins XR-11 X-ray machine, then treated with 20 µg/mL caffeine (cat#C0750, Merck Millipore, Burlington, MA, USA) and 10 µg/mL colchicine (cat#D00138122, Calbiochem, San Diego, CA, USA), then incubated for 6 h at 37 °C at 5% CO_2_. Cells were trypsinized and harvested with centrifugation (1000*× g*, RT, 3 min) and incubated for 30 min at 37 °C with 75 mM KCl. Cells were washed three times with ice-cold methanol and acetic acid in a 3:1 ratio, 20 µL was then dried onto coverslips and stained with Hoechst (1 μg/mL in PBS) before 150 chromosome spreads were counted and classified per sample.

### 2.12. Double Transfection of siRNAs and Transgenic Constructs

Hela cells were initially plated in a 6-well tissue culture plate and co-transfected with siControl and Flag vector, or siCNTRB along with Flag vector, or wild-type (WT), S781A, S781D or ARVA variants of Flag-CNTRB^180-903aa^. After 24 h, the cells were divided into two tissue culture plates. At 48 h post-transfection, half of the cells were subjected to X-ray treatment. Following X-ray treatment, the cells were treated with 20 µg/mL caffeine and 10 µg/mL colchicine and incubated for 6 h at 37 °C in a 5% CO_2_ environment. Chromosomes were prepared as described above.

### 2.13. Immunofluorescence and Microscopy

Twenty-four hours post-transfection, cells were seeded onto coverslips, incubated for another 24 h followed by fixation with 3% PFA for a duration of 10 min. Subsequently, the cells were permeabilized using 0.5% Triton X-100 in PBS for 10 min, then washed three times with PBS. To block non-specific binding, the cells were incubated in a blocking buffer containing 3% bovine serum albumin and 0.1% Triton X-100 in PBS for 1 h at room temperature (RT). The samples were then treated with primary antibody diluted in blocking buffer overnight at 4 °C. After incubation with the primary antibody, the cells were washed three times with 0.1% Triton X-100 in PBS and incubated with a fluorescently tagged secondary antibody diluted in blocking buffer for 1 h at RT in the dark. Following two washes with 0.1% Triton X-100 in PBS, the cells were counterstained with Hoechst (1 μg/mL in PBS) for 10 min. Microscopy images were captured using a Zeiss LSM800 system.

### 2.14. Antibodies

For immunoblotting (IB) anti-FlagM_2_ (IB: 1:10,000, cat#F1804, Merck Millipore, Burlington, MA, USA), anti-GFP (IB: 1:1000, cat#11814460001, Roche, Basel, Switzerland), anti-CNTRB (IB: 1:1000, cat#PA5-96990, Invitrogen, Waltham, MA, USA), anti-R3A (IB: 1:1250, cat#PA566644, Invitrogen, Waltham, MA, USA), anti-R3B (IB: 1:1250, cat#PA551545, Invitrogen), anti-PP4c (IB: 1:1000, cat#MA5-32946, Invitrogen), and anti-αTubulin (IB: 1:10,000, cat#T6199, Merck Millipore, Burlington, MA, USA) were used. For immunostaining, we used anti-γH2AX (IF: 1:500, cat#ab81299, Waltham, MA, USA), anti-γTubulin (IF: 1:300, cat#T6557, Merck Millipore, Burlington, MA, USA) and anti-flag (IF: 1:500, cat#F7425, Merck Millipore, Burlington, MA, USA). Secondary antibodies were the following: goat anti-mouse IgG conjugated to horseradish peroxidase (IB: 1:10,000, cat#P044701-2 Dako, Glostrup, Denmark), donkey anti-rabbit IgG Alexa Fluor 488 (IF: 1:500, cat#A21206, Thermo Fisher Scientific, Waltham, MA, USA) and donkey anti-mouse IgG Alexa Fluor 488 (IF: 1:500, cat#A21202, Thermo Fisher Scientific, Waltham, MA, USA), donkey anti-rabbit IgG Alexa Fluor 594 (IF: 1:500, cat#A21203, Thermo Fisher Scientific, Waltham, MA, USA).

### 2.15. Apoptosis Assay

HeLa cells were grown in 6-well tissue culture plates and transfected with gene-specific siRNAs to silence *cntrob*, *pp4c*, *r3a* or *r3b*, respectively. At 48 h post-transfection, cells were exposed to 2 Gy X-ray irradiation. At 8 h post-irradiation, cells were trypsinized, fixed with 70% ethanol on ice, and DNA was stained with 50 μg/mL propidium iodide (cat#P4170, Sigma) solution for 30 min followed by flow cytometric analysis (CytoFlexBecton, Dickinson and Company Biosciences, San Jose, CA, USA).

## 3. Results

### 3.1. The R3 Subunit of PP4 Directly Binds to CNTRB through Its FRVP Motif

It has been reported that the carboxy-terminal half of human CNTRB (CNTRB^460-903aa^), which contains a genuine PP4-binding motif (771-FRVP-774, Figure 1a), specifically interacts with both isoforms of R3 in HeLa cells [29]. Therefore, it was critical to understand whether the binding of full-length CNTRB to R3A or R3B depends exclusively on this single FRVP motif or if additional sequences are also needed for the interaction. To test this, we performed an in vitro pull-down assay and found that immobilized recombinant GST-R3A and GST-R3B specifically bind to the ^35^S-methionine-labeled wild-type CNTRB produced in in vitro-coupled transcription and translation reactions (IVTT). However, when we mutated phenylalanine and proline to alanine in the FRVP motif (**F**RV**P** to **A**RV**A**), the interaction was completely abolished (Figure 1b). This indicates that CNTRB has a single PP4-binding motif and that FRVP is necessary and sufficient for R3A/R3B recruitment.

We encountered difficulties when we wanted to validate the in vitro binding results with in vivo experiments using full-length CNTRB. We (Appendix A), and others [32], have found that overexpressed CNTRB forms aggregates around the nucleus and on the cytoskeleton and does not localize to the centrosome. Moreover, we observed the same when the amino-terminal (CNTRB^1-460aa^) or carboxy-terminal (CNTRB^460-903aa^) [29] halves of CNTRB were overexpressed separately in human cells (Appendix A). This may be due to the otherwise low levels of endogenous CNTRB, which is tightly regulated, as well as due to interrupted structural elements in the truncated overexpressed proteins (Appendix A). Therefore, based on secondary structure prediction, we designed a transgenic fragment of CNTRB lacking the first 180 amino acids (hereafter CNTRB^180-903aa^). When overexpressed, CNTRB^180-903aa^ was soluble and showed centrosomal localization in human cells (Appendix A). Therefore, we co-expressed flag-tagged CNTRB^180-903aa^ and its PP4-binding-deficient mutant, ARVA, together with GFP, GFP-R3A, or GFP-R3B, respectively, in HEK293 cells. Using co-immunoprecipitation experiments, we demonstrated that both GFP-R3A and GFP-R3B bind to CNTRB^180-903aa^, which requires an intact FRVP motif (Figure 1c). All these results imply that PP4 binds specifically to a single motif in CNTRB^180-903aa^. In addition, we proved that CNTRB^180-903aa^ is a soluble centrosomal protein, with the potential for use in functional assays.

### 3.2. CNTRB and PP4 Co-Operate in DNA Damage Repair

The variant of histone H2AX undergoes rapid phosphorylation at serine 139 (referred to as γH2AX [48]) in the microenvironment of chromatin surrounding DSBs [49]. This modification facilitates the accumulation of DNA repair machinery [50] and acts as a signal for DNA damage by activating other DNA repair factors [51]. As mentioned earlier, PP4 is responsible for dephosphorylation of γH2AX, which is necessary to restore the normal progression of the cell cycle after DNA damage has been repaired [20]. To explore the link between CNTRB and PP4 during DNA repair in the S-phase, we utilized EdU (5-ethynyl 2′-deoxyuridine) pulse-labeled HeLa cells and compared the dynamics of γH2AX in cells silenced for *cntrob*, *pp4c*, *r3a*, *r3b*, or *cntrob* and *pp4c*, respectively, after X-ray irradiation. We quantified the downregulation of the indicated genes by western blotting and qPCR (Appendix A and Appendix A). All conditions showed a significant increase in the number of γH2AX nuclear foci compared to the control sample, indicating a synergistic connection between CNTRB and PP4 function during the DNA repair process (Figure 2a and Appendix A). Sustained levels of γH2AX 8 h after irradiation suggest a strong delay in DNA repair (Figure 2a, Appendix A). To validate these results, we repeated the experiment with a second set of siRNAs with different target sequences and found a similar phenotypic change in cells (Appendix A and Appendix A). Simultaneously, we quantified the downregulation of the indicated genes by qPCR, normalized to actin control (Appendix A and Appendix A).

Next, we measured the frequency of HR applying the commonly used DR-GFP U2OS reporter cell line. The reporter cell line carries two truncated *gfp* (green fluorescent protein-encoding) cassettes: the first fragment contains an I-SceI rare cutter nuclease recognition site, and the second fragment contains the missing 466 bp to prevent GFP expression by HR action. In detail, when I-SceI cuts and generates the DSB, homologous recombination takes place using the other truncated *gfp* (*igfp*) as a template to restore the GFP cassette, resulting in functional GFP. In the end, the proportion of the GFP signal-carrying cells reflects the efficiency of the HR. To apply this approach, we detected a strong decrease in HR when we downregulated *cntrob*, *pp4c*, *r3a*, *r3b* separately, or *cntrob* and *pp4c* together (Figure 2b and Appendix A). These results confirm our previous findings and suggest that PP4 cooperates with CNTRB in the regulation of HR.

To study the effect of CNTRB, PP4c, R3A, R3B, or CNTRB and PP4 depletions in DSB repair at the chromosome level, we isolated chromosome spreads after X-ray irradiation. We observed an increased number of chromosome aberrations, particularly a phenotype characterized by closed chromosome arms. One possible explanation for this observed phenotype is that the Holliday junction structures, which form during DNA repair triggered by X-ray exposure, were inadequately resolved. As a consequence, the sister chromatids were retained instead of being separated properly. The frequency of this phenotype was significantly higher in the CNTRB- or PP4-silenced cells compared to the control group (Figure 2c,d and Appendix A), suggesting that the depletion of these transcripts leads to a defect in DSB repair. Furthermore, we assessed the proportion of apoptotic cells during our examination. However, we did not observe a notably elevated fraction of apoptotic cells. As a benchmark, we evaluated H_2_O_2_-treatment, and in this case, we identified a substantial increase (Appendix A).

Taken together, these results indicate that CNTRB and PP4c regulate the resolution of Holliday junction structures through cooperation in HR-mediated DNA repair.

### 3.3. Role of SQ and FRVP Motifs of CNTRB and Their Associated Phenotypes in the DNA Damage Response

A SILAC (stable isotope labeling with amino acids in cell culture) study has revealed that Ser781 in the SQ motif of CNTRB is phosphorylated by either ATM or ATR kinase [39]. Recently, it was shown that the ATR kinase phosphorylates CNTRB, which is required for DNA repair by HR. However, the reversing phosphatase remains unknown [40,53]. We note that this particular SQ motif is located adjacent to the PP4-binding site (771-FRVP-774 of CNTRB, Figure 1a), which is specifically recognized by R3A and R3B subunits of PP4 (Figure 1b,c) [29]. To investigate the importance of the SQ motif (in fact, Ser781), as well as the PP4-recognition FRVP sequence in CNTRB function during DNA damage response, we generated mutated forms of CNTRB with the following amino acid substitutions: Ser781 in the SQ motif was replaced with alanine (S781A) to generate a non-phosphorylatable form, or with aspartate (S781D) to create a phosphor-mimetic form of CNTRB; and the phenylalanine (F) and proline (P) in the FRVP motif were replaced with alanine (FRVP to ARVA) to abolish the PP4 interaction (Figure 1b,c) [29]. We first assessed the binding capacity of recombinant GST, GST-R3A, and GST-R3B to the ^35^S-labeled full-length CNTRB-S781A or CNTRB-S781D using the GST-IVTT in vitro binding assay [43]. Interestingly, none of the mutations affected the interaction (Figure 3a). We then performed an in vivo co-immunoprecipitation experiment, which showed that both GFP-R3A and GFP-R3B bound to the S781A and S781D variants of Flag-CNTRB^180-903aa^ (Figure 3a), similarly to the in vitro direct binding (Figure 3b). These results indicate that the interaction between CNTRB and R3A or R3B requires the intact FRVP PP4-binding motif (Figure 1b,c) but is likely independent of the SQ motif of CNTRB (Figure 3a,b).

Next, we tested whether these CNTRB variants have an impact on the modulation of DNA repair progression. We co-transfected cells with Flag-tagged wild-type CNTRB^180-903aa^ or its ARVA, S781A or S781D derivatives, respectively, along with CNTRB siRNA. It is important to note that the siRNA we used exclusively targeted the endogenous *centrob* transcript (its 5′ end, which is missing from the Flag-tagged mutants, Appendix A), and did not affect the levels of the transgenic proteins (Appendix A). This allowed us to specifically assess the function of the indicated Flag-CNTRB derivatives. Using this experimental setup, we performed a chromosome aberration assay and observed an increased number of abnormal closed-arm morphology [5] in the siCNTRB sample upon X-ray irradiation, which was restored in cells expressing wild-type CNTRB fragment (Figure 3c,d and Appendix A). However, the expression of the ARVA, S781A, or S781D mutation-containing CNTRB derivatives did not restore the function of CNTRB, suggesting that PP4 binding to CNTRB, as well as the integrity of the SQ motif, is key in the regulation of DNA repair (Figure 3c,d).

To summarize, our results emphasize the importance of PP4 binding to the single FRVP motif, as well as the ATR-targeted SQ motif, in the functioning of CNTRB, particularly in its regulation of DNA repair progression in response to DNA damage.

## 4. Discussion

Understanding the molecular etiology of cancer plays an indispensable role in shaping the future of cancer diagnosis and personalized therapy for patients. The Ser/Thr protein phosphatase 4, PP4, has been extensively studied and found to play a critical role in several cellular processes, including cell cycle regulation [15,24,30,31,54,55,56,57,58,59,60] and DNA repair [16,17,19,61,62,63,64]. The impact of PP4 on cancer development has been investigated, revealing that alterations in PP4 expression, subcellular localization, or activity can contribute to tumorigenesis. In breast, lung, colon, and prostate cancer, an increase in PP4 expression is often observed, suggesting a possible oncogenic role of this phosphatase [65,66,67]. On the other hand, a marked decrease in PP4 expression or loss of PP4 activity is observed in ovarian and cervical cancer, highlighting its tumor suppressive functions [68]. Moreover, there is compelling evidence demonstrating the role of PP4 in DNA repair through its regulation and termination of signaling events [13]. A specific target of PP4 is γH2AX, which becomes phosphorylated upon DNA damage. The dephosphorylation of γH2AX by PP4 is essential for restoring the normal progression of the cell cycle after the completion of DNA repair [20].

Centrosomes are multifunctional regulators of genome stability and ultimately serve as scaffolds for DNA repair proteins [69]. Centrosomes are the major microtubule organizing centers in animal cells and modulate the dynamics of microtubules, too, which are required for DDR protein transportation when DNA is damaged [53,70]. It has recently been shown that the daughter centriole-specific Centrobin (CNTRB), which is needed for centrosome duplication [32] and microtubule stability (regulated by the activity of NEK2 and polo-like kinase 1) [36,37,38], also localizes to DSB sites and interacts with important HR factors, indicating its direct involvement in the repair process [40]. CNTRB has been identified as a BRCA2-interacting protein, one of the major regulators of DNA repair by HR [32]. In addition, CNTRB has been found to be phosphorylated by the ATR/ATM kinase, further supporting its role in DNA repair [39,40]. We, and others, have shown that Centrobin is a genuine target of PP4 [29,30]; however, whether PP4 is involved in the DDR function of CNTRB has not yet been studied.

Through rigorous experiments, we have successfully confirmed a direct interaction between the regulatory subunits of PP4 phosphatase R3A or R3B, and full-length CNTRB (Figure 1). We provided evidence that the interaction depends on the presence of a single PP4-binding FxxP motif (FRVP) within CNTRB, with any alterations or mutations in the phenylalanine (F) and proline (P) residues potentially disrupting the binding capacity. In addition, our research has shown that knocking down CNTRB, along with members of the PP4 complex (including the catalytic subunit PP4c, and the substrate-binding subunit R3A or R3B), leads to a reduction in homologous recombination frequency (Figure 2b). This suggests that both CNTRB and the PP4 complex are actively involved in the same signaling pathway, highlighting their interconnectedness. Furthermore, our data suggest that CNTRB, as a target of the PP4 holoenzyme, has a similar influence on the status of the histone variant H2AX, which plays a crucial role in the DNA repair response (Figure 2a). Remarkably, when we selectively silenced *cntrob*, *pp4c*, *r3a*, *r3b*, or the combined knockdown of *pp4c* and *cntrob*, we observed a significant increase in a chromosome aberration called “arms closed” morphology (Figure 2c). This phenotype is likely due to incomplete Holliday junction resolution. Furthermore, CNTRB phosphorylated by ATR is required for both cell survival and homologous recombination after induction of DNA damage [40]. We showed that expression of ARVA (deficient in PP4-binding), and S781A or S781D (phospho-null or phospho-mimetic forms of Ser781 in the SQ motif) variants of CNTRB could not rescue the CNTRB-silencing induced chromosome aberration phenotype in gamma-irradiated cells (Figure 3c,d) compared to wild-type CNTRB. This finding suggests that the putative phosphorylation site within the SQ motif and the PP4 consensus recognition motif, FRVP, of CNTRB are crucial in DNA repair, most probably by regulating the function of CNTRB in the resolution of the Holliday structures. Based on literature data and our results, we assume that the phosphoregulation of CNTRB might contribute to the interaction between CNTRB and BRCA2, thereby promoting the process of homologous recombination. Notably, the precise characterization of this interaction remains the focus of future research endeavors. Considering our comprehensive results, it is evident that the PP4 holoenzyme and CNTRB jointly contribute to DNA repair by homologous recombination.

In summary, our research emphasizes the interplay between PP4 and CNTRB in the context of genome instability and DNA repair. By elucidating the molecular mechanisms underlying their interactions and their roles in homologous recombination, these findings support potential therapeutic targets and treatment strategies.

## 5. Conclusions

In this research, we present supporting evidence for the physical interaction between PP4 and CNTRB. Additionally, we illustrate the essential role played by the intact FRVP and SQ motifs within CNTRB in mending double-stranded DNA breaks resulting from X-ray irradiation. These breaks are repaired through human cell homologous recombination (HR) mechanisms. When PP4 is excluded from CNTRB or mutations affect the SQ motif, an HR delay ensues. This delay leads to the accumulation of chromosomes displaying irregular morphology, likely attributable to the impaired resolution of Holliday junctions. Our discovery proposes a novel role for PP4 during the later stages of HR, further validating its participation in facilitating the resumption of the cell cycle post-DNA repair procedures.

## Figures and Tables

**Figure 1 cells-12-02219-f001:**
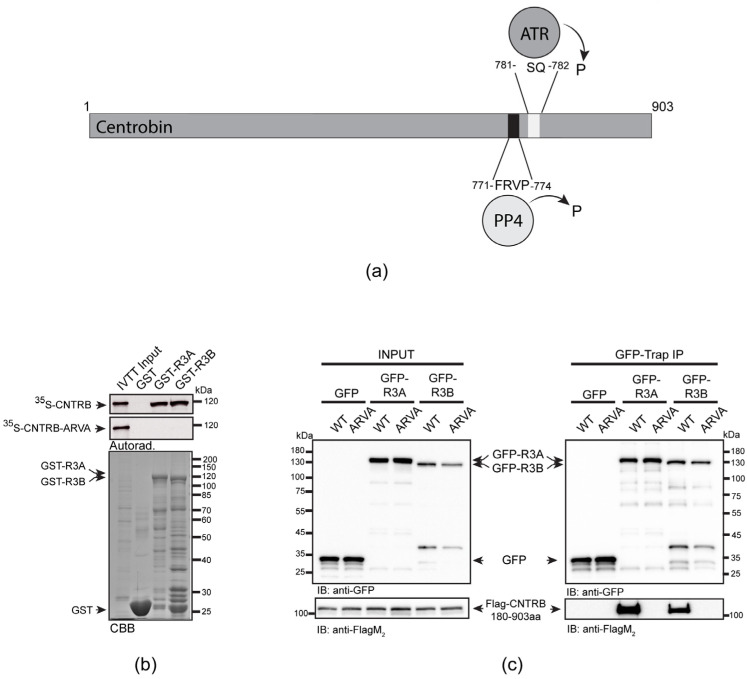
R3A and R3B bind directly to CNTRB through the PP4-binding motif. (**a**) Schematic representation of the human CNTRB protein: the PP4-binding FRVP sequence and the ATR kinase-recognition SQ motif are in proximity. Numbers indicate amino acid positions. P indicates putative (de)phosphorylation. (**b**) Autoradiogram (Autorad.) of GST-IVTT in vitro binding assay demonstrates that GST-R3A and GST-R3B (baits) specifically bind to ^35^S-methionine-labeled CNTRB (^35^S-CNTRB) produced in IVTT. The interaction requires an intact FRVP motif in CNTRB, because the ARVA mutant (^35^S-CNTRB-ARVA) does not bind to the bait proteins. The Coomassie brilliant blue-stained gel (CBB) represents the loading of the bait proteins. GST serves as a negative control bait. (**c**) GFP-trap co-immunoprecipitation of GFP (negative control), GFP-R3A or GFP-R3B transiently co-expressed with either Flag-CNTRB^180-903aa^ (indicated as WT) or Flag-CNTRB^180-903aa^-ARVA (indicated as ARVA) in HEK293 cultured cells show that CNTRB binding to R3A or R3B requires the intact FRVP motif in vivo. Cell lysate inputs and purified proteins were subjected to SDS-PAGE and western blot analysis using the indicated antibodies. Ponceau S-stained membranes are shown in Appendix A.

**Figure 2 cells-12-02219-f002:**
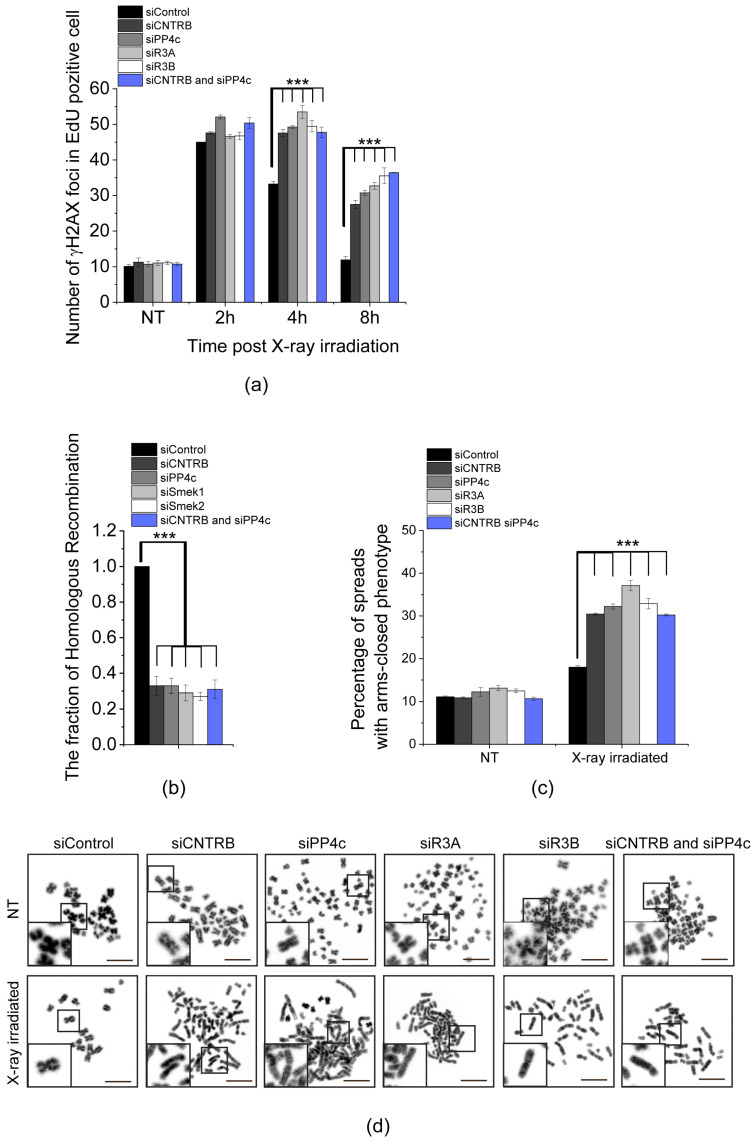
PP4 and CNTRB act together during DNA damage response. (**a**) Enumeration of γH2AX foci in HeLa cells. Cells were transfected with the following siRNAs: siControl, siCNTRB, siPP4c, siR3A, siR3B, or co-transfected with siCNTRB and siPP4c. After 48 h of RNAi, DNA damage was induced with 2 Gy X-ray irradiation and γH2AX foci were counted in 50 cells at different times (2, 4, and 8 h) post-irradiation. NT indicates the non-treated (non-irradiated) samples. Graphs include all data points and mean ± SEM (*n* = 3, three replicates). Asterisks indicate *p*-values (*** *p* < 0.001) obtained by linear regression fitted independently for each time point. (**b**) Homologous recombination measuring assay. Cells were transfected with the following siRNAs: siControl, siCNTRB, siPP4c, siR3A, siR3B, or co-transfected with siCNTRB and siPP4c. After 48 h, the cells were transfected with I-SceI expressing plasmid construct. The number of GFP-positive cells was calculated with Cellprofiler [52] and normalized to siControl-transfected cells. Graphs include all data points and mean ± SEM (*n* = 3, three replicates). Asterisks indicate *p*-values (*** *p* < 0.001) obtained by linear regression fitted independently for each time point. (**c**,**d**) Chromosome study in CNTRB- or PP4 subunit-depleted cells. Cells were transfected with the following siRNAs: siControl, siCNTRB, siPP4c, siR3A, siR3B, or co-transfected with siCNTRB and siPP4c. After 48 h, the cells were treated with 2 Gy X-ray irradiation and treated for 6 h with 20 µg/mL caffeine and 10 µg/mL colchicine. NT refers to non-treated (non-irradiated) cells. (**c**) Chromosome aberrations were counted in 150 spreads in each sample, and the percentage of arms closed phenotype was plotted. Graphs include all data points and mean ± SEM (*n* = 3, three replicates). Asterisks indicate *p*-values (*** *p* < 0.001) obtained by linear regression fitted independently for each time point. (**d**) Representative images of chromosome spreads isolated from the indicated cells in NT or in X-ray irradiated samples. Scale bar is 10 µm. Insets at bottom left show magnified representative chromosome morphologies.

**Figure 3 cells-12-02219-f003:**
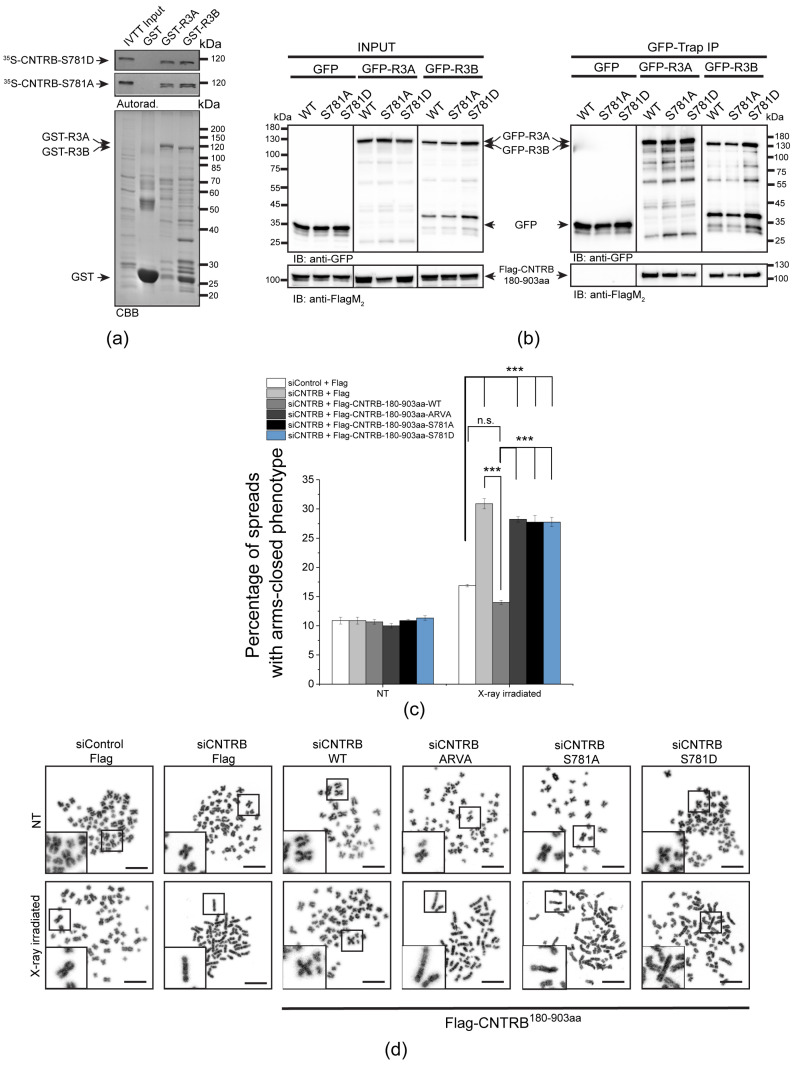
The FRVP and SQ motifs are crucial for CNTRB-regulated DNA repair. (**a**) Autoradiogram (autorad.) of GST-IVTT in vitro binding assay demonstrates that GST-R3A and GST-R3B (baits) interact with the S781A and S781D mutants of ^35^S-methionine-labeled CNTRB (^35^S-CNTRB-S781A and ^35^S-CNTRB-S781D) produced in IVTT. The Coomassie brilliant blue-stained gel (CBB) represents the loading of the bait proteins. GST serves as a negative control bait. (**b**) GFP-trap co-immunoprecipitation of GFP (negative control), GFP-R3A or GFP-R3B transiently co-expressed with either Flag-CNTRB^180-903aa^ (indicated as WT), Flag-CNTRB^180-903aa^-S781A (indicated as S781A), or Flag-CNTRB^180-903aa^-S781D (indicated as S781D) in HEK293 cultured cells shows that CNTRB binding to R3A or R3B is independent of the integrity of the SQ motif in vivo. Cell lysate inputs and purified proteins were subjected to SDS-PAGE and western blot analysis using the indicated antibodies. Ponceau S-stained membranes are shown in Appendix A. (**c**,**d**) Chromosome study in siControl or siCNTRB cells co-transfected with empty Flag plasmid or CNTRB-depleted (siCNTRB) cells co-transfected with the wild-type or S781A/S781D/ARVA mutant variants of Flag-CNTRB^180-903aa^. After 48 h, the cells were treated with 2 Gy X-ray irradiation and treated for 6 h with 20 µg/mL caffeine and 10 µg/mL colchicine. NT refers to non-treated (non-irradiated) cells. (**c**) Chromosome aberrations were counted in 150 spreads in each sample, and the percentage of arms closed phenotype was plotted. Graphs include all data points and mean ± SEM (*n* = 3, three replicates). Asterisks indicate *p*-values (*** *p* < 0.001) obtained by linear regression fitted independently for each time point. n.s.: not significant. (**d**) Representative micrographs of chromosome spreads isolated from cells after X-ray irradiation or without treatment (NT). Scale bar is 10 µm. Insets at bottom left show magnified representative chromosome morphologies.

## Data Availability

Not applicable.

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
