# Peer review of "Protein Phosphatase 4 Is Required for Centrobin Function in DNA Damage Repair"

_cells, 2023, doi:10.3390/cells12182219_

Round 1

Reviewer 1 Report

This manuscript has investigated the interaction between PP4 and centrobin, and the significance in DNA damage repair. They found PP4 binds centrobin independent of its phosphorylation by ATR kinase, but depending on a PP4-binding FRVP sequence. Depletion of PP4 or centrobin leads to reduced homologous recombination and aberrant chromosomes, which cannot be rescued by centrobin mutant that doesn’t bind PP4. The manuscript has revealed the function of PP4 in DNA damage repair and the possible pathway through centrobin, and should be interesting to the field. The overall experiments have been carefully carried out and the results are solid. I would support its publication in Cells once some minor points are answered.

1.     The short name for centrobin, according to NCBI, is CNTRB (protein), or CNTROB (gene). Otherwise, Drosophila centrobin is named Cnb.

2.     The interactions between R3A/R3B and centrobin phosphor-mutant show certain change compared to centrobin WT. The author should perform at least three repeats and show the statistic result.

Author Response

Please see the attachmen.

Reviewer 2 Report

The present study provides some mechanistic insights regarding the interaction between Ctb and PP4 through their binding domains in cycling cells and this interaction is likely to be required for the HR pathway. The study suffers from direct evidence that the Ctb-PP4 axis is essential for successful HR completion through its interactions with DDR and HR factors. The authors may consider following the below suggestions and revising the manuscript to be accepted in the future. 

1) Please provide representative gammaH2AX foci images for all time points and treatment conditions related to Figure 2a.

2) Please provide IB images for all downregulated factors using respective siRNA. The authors may consider including them in the main figure itself.

3) Please perform co-IP assay using gammaH2AX and/or Rad51 to show the interaction modulation of Ctb or PPR with or without mutations in their binding domains.

4) Perform an apoptosis assay under described treatment conditions to show the importance of the Ctb-PP4 interaction in cell survival after DNA damage induction.

The English language style is appropriate. There are no major grammatical mistakes detected. 

Author Response

Please see the attachmen.
